# Bacopasides I and II Act in Synergy to Inhibit the Growth, Migration and Invasion of Breast Cancer Cell Lines

**DOI:** 10.3390/molecules24193539

**Published:** 2019-09-30

**Authors:** Helen M. Palethorpe, Eric Smith, Yoko Tomita, Maryam Nakhjavani, Andrea J. Yool, Timothy J. Price, Joanne P. Young, Amanda R. Townsend, Jennifer E. Hardingham

**Affiliations:** 1Solid Tumour Group, Basil Hetzel Institute for Translational Health Research, The Queen Elizabeth Hospital, Woodville South, SA 5011, Australia; helen.palethorpe@unisa.edu.au (H.M.P.); eric.smith@adelaide.edu.au (E.S.); yoko.tomita@sa.gov.au (Y.T.); maryam.nakhjavani@adelaide.edu.au (M.N.); timothy.price@sa.gov.au (T.J.P.); joanne.young@adelaide.edu.au (J.P.Y.); amanda.townsend@sa.gov.au (A.R.T.); 2Adelaide Medical School, University of Adelaide, Adelaide, SA 5005, Australia; andrea.yool@adelaide.edu.au; 3Medical Oncology, The Queen Elizabeth Hospital, Woodville South, SA 5011, Australia

**Keywords:** bacopaside I, bacopaside II, synergy, breast cancer, migration, spheroid invasion, triple negative breast cancer

## Abstract

Bacopaside (bac) I and II are triterpene saponins purified from the medicinal herb *Bacopa monnieri*. Previously, we showed that bac II reduced endothelial cell migration and tube formation and induced apoptosis in colorectal cancer cell lines. The aim of the current study was to examine the effects of treatment with combined doses of bac I and bac II using four cell lines representative of the breast cancer subtypes: triple negative (MDA-MB-231), estrogen receptor positive (T47D and MCF7) and human epidermal growth factor receptor 2 (HER2) positive (BT-474). Drug treatment outcome measures included cell viability, proliferation, cell cycle, apoptosis, migration, and invasion assays. Relationships were analysed by one- and two-way analysis of variance with Bonferroni post-hoc analysis. Combined doses of bac I and bac II, each below their half maximal inhibitory concentration (IC_50_), were synergistic and reduced the viability and proliferation of the four breast cancer cell lines. Cell loss occurred at the highest dose combinations and was associated with G2/M arrest and apoptosis. Migration in the scratch wound assay was significantly reduced at apoptosis-inducing combinations, but also at non-cytotoxic combinations, for MDA-MB-231 and T47D (*p* < 0.0001) and BT-474 (*p* = 0.0003). Non-cytotoxic combinations also significantly reduced spheroid invasion of MDA-MB-231 cells by up to 97% (*p* < 0.0001). Combining bac I and II below their IC_50_ reduced the viability, proliferation, and migration and invasiveness of breast cancer cell lines, suggesting synergy between bac I and II.

## 1. Introduction

Breast cancer remains the second most common cancer in women worldwide and the fourth leading cause of cancer death [1]. Breast cancer has been classified traditionally according to characteristics such as tumour size, lymph node involvement, histological grade and the receptors expressed: estrogen receptor (ER+) and/or progesterone receptor (PR+), human epidermal growth factor receptor 2 (HER2) amplified (referred to as HER2+), and triple negative breast cancer (TNBC), which lacks the expression of all three markers (ER and PR, and HER2) [2]. Endocrine therapy is the main initial therapy for ER+/PR+ breast cancer; however, inherent and acquired resistance leads to disease progression and incurable disease (reviewed in [3]). Similarly, whilst anti-HER2 therapies, such as trastuzumab in combination with chemotherapy, have significantly increased overall survival for patients with early and advanced HER2-amplified breast cancer, resistance mechanisms lead to recurrent and metastatic disease [4]. TNBC is unresponsive to endocrine or anti-HER2 therapies, since it lacks the expression of the necessary receptors. TNBC accounts for 10–15% of all diagnosed breast cancer and has the poorest prognosis due to its biologically aggressive nature. Treatment relies on surgery, chemotherapy and radiotherapy; however, the disease-free interval with neoadjuvant or adjuvant therapy is shorter and metastatic disease more aggressive than other types of breast cancer [5]. Despite a number of clinical trials in TNBC, drug treatment regimens are yet to be optimised, highlighting a need for additional novel drug therapies.

Terpenoids are a diverse family of natural products found in a variety of fruits, vegetables and medicinal plants [6]. Classified according to the number of cyclic structures that they contain, terpenoids are a major class of anticancer compounds. Perhaps the most well-known terpenoid in clinical use is paclitaxel, a diterpenoid used for the last 30 years to treat early and metastatic breast cancer as a first-line or second-line treatment, either alone or in combination with other chemotherapeutics [7]. Disadvantages of its use, however, include adverse effects and multi-drug resistance [8,9]. Another subgroup, triterpene saponins, are noted to have cytotoxic, anti-mutagenic, anti-inflammatory and anti-cancer activities (reviewed in [10]). However, limited studies have investigated the anti-cancer potential of the bacopasides, which are members of this subgroup, derived from the medicinal herb *Bacopa monnieri* [11,12,13,14], and no studies have been performed using the purified compounds bacopaside (bac) I or II. Use of the whole plant extract has been used in traditional Indian medicine for 3000 years, primarily to enhance memory and cognitive function [11,15], with no evidence of adverse reactions or side effects in humans (reviewed in [16]). We therefore aimed to explore the anti-cancer potential of bac I and bac II against breast cancer.

To further support the use of bac I and bac II in cancer, we previously showed that these drugs block the functional activity of the transmembrane protein aquaporin (AQP) 1 [12]. AQP1 is involved in ion and water transport across the cell membrane and is expressed in human tissues, including in renal tubules, intestinal epithelia, breast epithelia and vascular endothelial cells [17]. Bac I specifically blocks both the ion and water channel function of AQP1, while bac II blocks the water channel function only [12]. Several studies support a role for AQP1 in tumour progression: AQP1-null mice showed reduced tumour growth, metastasis and angiogenesis compared to AQP1-expressing mice [18,19]; mammary gland tumour cells transfected with AQP1 showed increased migration in vitro and increased tumour cell extravasation and lung metastases in vivo [20]; and in vitro inhibition of AQP1 reduced endothelial cell tube formation and the migration, invasion, and growth of colorectal cancer cell lines [12,13,14,21,22]. Clinically, AQP1 expression was increased in breast cancer compared to adjacent normal tissue [23,24,25], and high AQP1 expression was associated with TNBC and poorer progression-free and overall survival [24,26].

Together, these studies provide a strong rationale to investigate the effects of bac I and II in breast cancer. We therefore tested the anti-cancer activity of the drugs individually and combined on breast cancer cell lines from the major molecular subtypes, MDA-MB-231 (TNBC), T47D (ER+/PR+), MCF7 (ER+/PR−), and BT-474 (HER2+), and looked for potential synergy between the two drugs.

## 2. Results

### 2.1. Determination of the IC50 for Bacopasides I and II Individually and Combined

A viability assay (MTS) was performed on the four breast cancer cell lines treated with varying concentrations of either bac I or bac II. The cell lines showed a dose response effect to both drugs with half maximal inhibitory concentrations (IC_50_s) shown in Figure 1. The IC_50_ values for bac I for MDA-MB-231, T47D and MCF7 were 99 μM (95% confidence interval (CI) 91–109), 89 μM (95% CI 73–109), and 83 μM (95% CI 79–87), respectively, whilst BT-474 was more sensitive at 59 μM (95% CI 56–63) (Figure 1a). The IC_50_ values for bac II for MDA-MB-231, MCF7 and BT-474 were 18 μM (95% CI 15–20), 19 μM (95% CI 17–20) and 16 μM (95% CI 15–17), respectively, whilst T47D was less sensitive at 29 μM (95% CI 9–96) (Figure 1b).

We then tested the effect of combinations of bac I and II on MDA-MB-231 cell viability. The MDA-MB-231 was selected since it showed high sensitivity for the two drugs and is most clinically relevant given the lack of targeted therapies for triple negative breast cancer. Since bac II was the more potent drug, with an IC_50_ 5.5-fold lower than for bac I for MDA-MB-231, we used a constant dose of bac II well below the IC_50_—specifically, 2.5 μM—whilst the concentration of bac I was varied; this was done to determine the effect of bac II, at a dose that did not reduce viability, on the IC_50_ of bac I. For MDA-MB-231 cells, the IC_50_ of the combinations was determined as 13 μM bac I (95% CI 10–16 μM) + 2.5 μM bac II. Bac II shifted the IC_50_ for Bac I from 99 μM (95% CI 91–109) to 13 μM (95% CI 10–16) (Figure 1c), indicating a synergistic effect of the two drugs. Drug synergy was supported using the isobologram, which showed that the IC_50_ of the two drugs combined was positioned below the theoretical line with a combination index (CIx) of 0.270 (95% CI 0.240–0.301) (Figure 1d). Isobolograms for the four cell lines are shown in Appendix A, with the combination indices shown in Appendix A.

### 2.2. Bacopasides I and II Combined Reduced the Proliferation of Breast Cancer Cell Lines

We investigated the effect of bac I and bac II, alone and in combination, on the proliferation of each cell line using a crystal violet assay on days 0, 1, and 3 post treatment. Figure 2 shows the growth curves for each of the breast cancer cell lines treated with vehicle, bac I and/or bac II. Proliferation compared to vehicle was significantly inhibited with all doses for MDA-MB-231. The dose of 10 μM bac I + 5 μM bac II resulted in the highest level of inhibition for all cell lines tested (all *p* < 0.0001 relative to vehicle at day 3). Relative to day 0, a decrease in absorbance occurred at day 1 with this dose for all cell lines (*p* ˂ 0.0001), as well as with 10 μM bac I + 2.5 μM bac II (*p* < 0.0001) for MDA-MB-231 and BT-474 (Figure 2a,d) and 5 μM bac I + 5 μM bac II (*p* = 0.0002) for BT-474 (Figure 2d), indicative of cell loss.

### 2.3. Higher Combination Doses of Bacopasides I and II Induced G2/M Arrest

Based on the cell loss induced with combination doses, cell cycle analysis was performed, comparing vehicle and drug-treated cells. For each breast cancer cell line, there was an increase in the percentage of cells in G2/M with increasing doses of bac I and bac II. The dose combinations at which this occurred were cell-line dependent. Compared to the vehicle, with a dose of 5 μM bac I + 5 μM bac II, the proportion of cells in G2/M increased by 17% and 9% for MDA-MB-231 (Figure 3a,b), and BT-474 (Figure 3g,h), respectively, (*p* < 0.0001), and by 13% and 20% with 10 μM bac I + 5 μM bac II for T47D (Figure 3c,d), and MCF7 (Figure 3e,f), respectively (*p* < 0.0001). Sub-G1 events, suggestive of dead cells and/or debris, increased with increasing combinations of bac I and bac II, most notably for MDA-MB-231 and BT-474.

### 2.4. Bacopasides I and II Combined Altered the Morphology of Breast Cancer Cell Lines

The morphology of breast cancer cell lines treated with each drug alone and in combination was observed microscopically after three days in culture. Bac I and bac II, each at 5 μM, and bac I, at 10 μM, had no observable effects on morphology compared to the vehicle control, while 10 μM bac II induced small vacuoles in MDA-MB-231 only; therefore, the highest doses of each drug selected for use in combination were doses of ≤10 μM bac I and ≤5 μM bac II (Appendix A). Combined doses of bac I and bac II induced the formation of prominent intracellular vacuoles. These appeared at lower combinations for MDA-MB-231 and BT-474, indicating their higher sensitivity compared to T47D and MCF7. The 5 μM bac I + 5 μM bac II and 10 μM bac I + 5 μM bac II combinations were the most potent, inducing prominent and large vacuoles in the majority of cells and occasional shrunken and rounded cells reminiscent of apoptosis, particularly for MDA-MB-231 and BT-474. The MCF7 cell line was the most resistant, displaying only small vacuoles and fewer apoptotic cells at the highest combinations. The two drugs were more potent when combined, since for 5 μM bac I + 5 μM bac II changes were more prominent compared to either 10 μM bac I or bac II alone (Appendix A). Apoptosis was confirmed by annexin-V staining.

### 2.5. Combination Doses of Bacopasides I and II Increased Annexin-V Staining

Cells were treated with a vehicle at a dose that slowed proliferation but did not cause cell loss and one or more doses that induced cell loss. Figure 4 shows the total apoptotic cells and representative scatterplots for each cell line. There was a significant increase in total apoptotic cells (early and late) compared to the vehicle with increasing combinations of bac I and bac II for all four cell lines. MDA-MB-231 and BT-474 were the most sensitive, with apoptosis occurring from 10 μM bac I + 2.5 μM bac II (*p* < 0.001, *p* < 0.0001 respectively) (Figure 4a,b,g,h). The increase in apoptosis was consistent with the increase in sub-G1 events for MDA-MB-231 and BT-474 in Figure 3; however, there was a lack of sub-G1 events with 5 μM bac I + 5 μM bac II for T47D and MCF7, despite there being an increase in apoptotic cells. This may be explained by the fact that sub-G1 represents debris and dead cells, whereas the apoptosis assay detects the exposure of phosphatidylserine on the outer leaflet of the plasma membrane—a surface change common to apoptotic cells early in the process. An increase in apoptotic cells may therefore include cells in an early phase of cell death that are not yet detectable in sub-G1.

### 2.6. Combination Doses of Bacopasides I and II Reduced Migration of Breast Cancer Cell Lines

We assessed the effects of bac I and bac II on breast cancer cell migration, since AQP1 is known to be important in this process and these drugs target AQP1 function. Figure 5 shows the rate of wound closure with the vehicle compared to bac I and bac II alone and combined for each breast cancer cell line. There was a significant inhibition of wound closure with combination doses of bac I and bac II, with MDA-MB-231 and BT-474 being most sensitive. For MDA-MB-231 (Figure 5a), wound closure at 20 h was inhibited by 69% with 5 μM bac I + 5 μM bac II compared to the vehicle (P < 0.0001), with significant differences observed as early as 4 h. For T47D (Figure 5b), migration at 96 h was inhibited by 43% with a combination dose of 10 μM bac I + 5 μM bac II (*p* < 0.0001). MCF7 cells were more resistant (Figure 5c): an inhibition of migration of 23% at 72 h occurred with 10 μM bac I + 5 μM bac II (*p* < 0.0001). For BT-474 (Figure 5d), at dose combinations of 10 μM bac I + 2.5 μM bac II and 5 μM bac I + 5 μM bac II, there was no wound closure, compared to 47% with the vehicle, with loss of cell adherence, as early as 24 h, causing an increase in the size of the wound (*p* < 0.0001). Representative images of wound closure assays are shown in Appendix A. As expected, the combinations that induced apoptosis, as assessed by annexin-V staining, reduced migration. However, dose combinations that did not induce apoptosis also reduced migration significantly, namely 5 μM bac I + 2.5 μM bac II for MDA-MB-231 and BT-474 and 10 μM bac I + 2.5 μM bac II for T47D (Figure 6). This suggests the drugs target other mechanisms of cell migration as well as inducing apoptosis.

### 2.7. Combination Doses of Bacopasides I and II Reduced Invasion of MDA-MB-231

Of the four breast cancer cell lines tested, only MDA-MB-231 displayed an ability to invade into the surrounding matrix by forming spindle-like protrusions (Figure 6b). As expected, the combination of 5 μM bac I + 5 μM bac II, which induced apoptosis, significantly reduced invasion into the surrounding matrix by 98% at day 3 (*p* < 0.0001). Furthermore, the doses of 2.5 μM bac I + 2.5 μM bac II and 5 μM bac I + 2.5 μM bac II significantly reduced invasion, by 68% and 97%, respectively (*p* < 0.0001)—doses that did not induce apoptosis (Figure 6a,b). This suggests that non-cytotoxic dose combinations reduced spheroid invasion by mechanisms other than the induction of cell death.

### 2.8. Combination Doses of Bacopasides I and II Reduced Transcript Expression of AQP1 in MDA-MB-231 Cells

AQP1 transcript expression was significantly higher in MDA-MB-231 relative to the other cell lines (*p* < 0.0001), being very low in T47D and BT-474, and undetectable in MCF7 (data not shown). Cell lines expressing AQP1 transcript were treated for 24 h with a non-cytotoxic combination of bac I and bac II that reduced migration: specifically, 5 μM bac I + 2.5 μM bac II for MDA-MB-231 and BT-474, and 10 μM bac I + 2.5 μM bac II for T47D. Cells were also treated with the 5 μM bac I + 5 μM bac II combination that induced apoptosis. Compared to vehicle, AQP1 transcript expression in MDA-MB-231 cells was reduced with the non-cytotoxic dose (*p* = 0.004) and by 2.5-fold with the cytotoxic dose (*p* < 0.0001) (Figure 7a). For T47D and BT-474, there was no significant effect on AQP1 transcript expression (Figure 7b,c). These findings suggest that the reduced proliferation, migration and invasion for MDA-MB-231 could be partly associated with a reduction in AQP1 transcript expression, particularly since combined doses of bac I and II had a limited effect on MCF7 cells, for which AQP1 was not detected.

## 3. Discussion

The cell membrane permeabilisation effects of high concentrations of saponins lead to hemolysis and cytotoxicity. At lower doses, however, anti-cancer properties such as the inhibition of proliferation, induction of apoptosis and attenuation of invasiveness have been demonstrated (reviewed in [27]). Here, we have shown for the first time that the triterpene saponins bac I and bac II in combination act in synergy to reduce the viability, proliferation, migration, and invasion of breast cancer cell lines in vitro. This anti-cancer activity of bac I and bac II occurred at high dose combinations that induced G2/M arrest and apoptosis but also at lower dose combinations that did not induce apoptosis. These lower non-cytotoxic combinations of bac I and II significantly reduced the migration of all four cell lines, with MDA-MB-231 and BT-474 being the most sensitive, and significantly reduced the spheroid invasion of MDA-MB-231.

There are no studies investigating the anti-cancer effects of combining bac I and II. Previous studies have focused on the anti-cancer effects of either the whole plant extract, bacoside A (a mixture of saponins from *B. monnieri* containing approximately 20% bac II w/w [28]), or bac I and bac II alone. In vitro, *B. monnieri* extract was cytotoxic to human cancer cell lines of the colon, lung, cervix, and breast [29,30,31]; bacoside A to human kidney carcinoma [32] and glioblastoma [33] cell lines; and bac I and bac VII to prostate, glioma, ileocecal, lung and breast cancer cell lines [11]. Similarly, we have shown bac II to induce apoptosis in vitro in colorectal cancer (CRC) cell lines [14] and in endothelial cells, suggesting anti-angiogenic potential [13]. We have also shown reduced migration of CRC cell lines expressing high AQP1 with bac I and bac II alone [12]. In vivo studies in mice have shown the chemopreventive activity of bacoside A in chemically-induced hepatocellular carcinoma (HCC) [34] and of *B. monnieri* extract in skin carcinogenesis [35,36], as well as anti-cancer activity of the extracts bac I and bacoside A in mouse models of melanoma [36], sarcoma [11], and Ehrlich ascites carcinoma (EAC) [32], respectively.

These studies provide strong evidence to investigate the anti-cancer efficacy of bac I and bac II combined in breast cancer cell lines from each of three main molecular subtypes, as performed here for the first time. We found that all cell lines were sensitive to the synergistic effects of bac I and bac II combined, but particularly MDA-MB-231 and BT474, representative of the more aggressive subtypes of breast cancer of TNBC and HER2+, respectively. We also found that certain dose levels and combinations were cytotoxic, consistent with previous studies using the whole extract and its derivatives.

A novel aspect of our study, however, was the finding that lower doses of bac I and II combined, at doses that did not induce apoptosis, also had potent anti-cancer activity in vitro. Previously, we showed that non-cytotoxic concentrations of bac II alone inhibited endothelial cell tube formation—an in vitro measure of angiogenesis [13]—but otherwise, only one study has shown the reduced motility of prostate cancer cell line DU145 with non-cytotoxic concentrations of *B. monnieri* extract [27]. This promising finding suggests that these drugs may have the potential to slow cancer progression and prevent metastasis, without the cytotoxicity to normal cells associated with chemotherapy [37].

In terms of safety, the highest doses used in our in vitro study to induce apoptosis were 10 μM bac I and 5 μM bac II, equivalent in mice to approximately 7 mg/kg and 4 mg/kg, respectively, while the human equivalent dose [38] would be approximately 0.6 mg/kg and 0.3 mg/kg, respectively. Extracts of the dried herb typically contain bac I and bac II in a 1:1 ratio, depending on the extraction method [39,40]. Rat toxicity studies showed no toxicity of *B. monnieri* extract containing bac I and bac II at levels higher than these. A large single oral dose of 5000 mg/kg of whole extract, containing 51.5 mg/kg bac I and 91 mg/kg bac II, caused no acute toxicity in Sprague-Dawley rats; in addition, rats receiving up to 1500 mg/kg of whole extract daily for 270 days, containing 15.5 mg/kg bac I and 27.3 mg/kg bac II, showed no evidence of chronic toxicity [41]. Others have studied the pharmacokinetics of bac I [42] and clinical effects of bac I, but not bac II, with no adverse effects in rats receiving 30 mg/kg orally for 6 days [43] and mice 50 mg/kg orally for 7 days [43]—5 to 8 times more bac I than the highest dose of bac I used in the present study. However, other animal studies and human clinical trials have used *B. monnieri* whole extract at too low a dose to be comparable to the amounts of bac I and bac II used here [44,45,46,47]. In terms of translation to human trials, based on the guidelines of the U.S. Department of Health and Human Services, Food and Drug Administration (FDA) [48], the estimation of the maximum safe starting dose in initial clinical trials is 10% of the predicted therapeutic dose, so for bac I and bac II in adults, this would be doses of 0.06 and 0.03 mg/kg, respectively, which is equivalent to approximately 4.2 mg or 2 mg, respectively, in an average 70 kg human. Even at the predicted therapeutic doses of 42 mg Bac I or 20 mg Bac II, these doses are much lower in comparison with other commonly used agents for metastatic breast cancer.

The rationale for combining the two drugs was based on their ability to block AQP1 ion and water channel function. Both bac I and bac II block the water channel function of AQP1, but only bac I blocks the ion channel function [12]. We therefore hypothesised their anti-cancer potential when combined, given that separately we have shown they reduced the migration of CRC cell lines that expressed high levels of AQP1 [12]. We found that the MDA-MB-231 cell line expressed the highest level of AQP1 transcript of the four breast cancer cell lines tested. This is consistent with a study showing relatively high AQP1 expression in TNBC [26]. Additionally, we found that, for the MDA-MB-231 cell line, AQP1 transcript expression could be reduced by bac I and II combined. This suggests the effects of the combined drugs might, in part, be mediated through AQP1. A limitation of our study, however, is the lack of AQP1 silencing or knockout experiments to confirm the role of AQP1 in the observations, which will be investigated in future studies.

There have been reports of other suggested mechanisms of anti-cancer activity for extracts of *B. monnieri* and bacoside A, the latter of which contains bac II but not bac I. Both induced the death of human glioblastoma cell lines in vitro through macropinocytosis, a process whereby cell membrane buckling leads to the accumulation of extracellular fluid inside large vacuoles within the cell [33]. This build-up of fluid within the cells might be as a result of the inhibition of AQP1 channel activity and hence inhibition of water flux. The morphological changes described in glioblastoma cells are consistent with the vacuolisation we observed in breast cancer cell lines with combined bac I and bac II treatment. In addition, the bacoside A co-treatment of male Wistar albino rats treated with N-nitrosodiethylamine (DEN) to induce hepatocellular carcinoma (HCC) reduced serum marker enzymes that are markedly elevated in HCC and increased the level of enzymatic and non-enzymatic antioxidants to near normal, thereby reducing disease-associated hepatic damage [34]. They also reported that bacoside A attenuated the increased expression of matrix metalloproteinase (MMP)-2 and -9 in DEN-induced HCC. Since MMPs are involved in tumour cell invasion and metastasis, an anti-metastatic effect of bacoside A was suggested [49]. The potential of combined doses of bac I and II to act via similar mechanisms in breast cancer should therefore be explored.

## 4. Materials and Methods

### 4.1. Reagents and Cell Lines

The analytical standards bacopaside I (CAS No. 382148-47-2, 89.6% purity by HPLC, Lot no. 00002002-T17H) and bacopaside II (CAS No. 382146-66-9, 98% purity HPLC, Lot Number: 00002002-T17H), derived from the medicinal herb *bacopa monnieri,* were obtained from ChromaDex (Irvine, CA, USA), solubilised in methanol at 10 mM and 1.5 mM stock solutions, respectively, and stored at −20 °C. Cell lines MDA-MB-231, T47D, MCF7 and BT-474 were purchased from the American Type Culture Collection (ATCC; Manassas, VA, USA). Cells were maintained in complete medium, either in Dulbecco’s modified Eagle’s medium (DMEM; Life Technologies, Carlsbad, CA, USA) for MDA-MB-231, MCF7 and BT-474 or Roswell Park Memorial Institute (RPMI) 1640 medium (Life Technologies, Carlsbad, CA, USA) for T47D, containing 10% foetal bovine serum (FBS) (Corning, NY, USA), 200 U/mL of penicillin, 200 μg/mL of streptomycin (pen strep; Life Technologies, Carlsbad, CA, USA) and 2 mM L-alanyl-L-glutamine dipeptide (GlutaMAX Supplement; Life Technologies, Carlsbad, CA, USA). Cells were grown under standard culture conditions at 37° C with 5% CO_2_ in air and used within 4 passages.

### 4.2. Cell Viability Assay for Determining IC50 and Drug Synergy

The effect of bac I and bac II, alone and in combination, on cell viability was determined by MTS assay as described previously [13,21]. Cells were seeded in complete medium at 1 × 10^4^ cells per well of a 96 well plate and incubated under standard culture conditions overnight. Next, cells were treated with either vehicle (1% methanol), bac I, bac II or combinations of bac I and bac II, for 24 h. MTS assay was performed using the CellTiter 96 AQueous Non-Radioactive Cell Proliferation Assay (Promega, Madison, WI, USA) according to the manufacturer’s instructions. Absorbance was read at 492 nm and results were calculated as the mean absorbance normalized to the vehicle control. The half maximal inhibitory concentration (IC_50_) for each drug alone and in combination was determined by non-linear regression analysis.

To determine the IC_50_ of bac I and bac II combined, bac II was used at a constant concentration of 2.5 μM whilst the concentration of bac I was varied. The isobologram method [50,51] was used to produce a theoretical line by plotting the IC_50_s for bac I and bac II alone on the *y*- and *x*-axis respectively. The IC_50_ combination of bac I and bac II was also plotted on the same graph. Its position would indicate whether the IC_50_ combination was antagonistic, additive, or synergistic, depending on whether the point was above, on or below the theoretical line respectively. This was also reflected by the combination index, calculated by the formula, CI = d1/D_x_1 + d2/D_x_2, where D_x_1 was the IC_50_ of bac I, D_x_2 was the IC_50_ of bac II, and d1 and d2 were the doses of bac I and bac II respectively that combined gave an IC_50_. The combination was antagonistic, additive or synergistic if the combination index was greater than, equal to or less than one [50].

### 4.3. Cell Proliferation

The crystal violet assay was used to measure cell proliferation and was performed as described previously [52]. Briefly, cells were seeded in complete medium at 2 × 10^3^ (MDA-MB-231 and MCF7), 3 × 10^3^ (T47D) or 4 × 10^3^ (BT-474) cells per well of a 96 well plate followed by overnight incubation. Cells were then treated with the vehicle or bac I and/or bac II, and crystal violet absorbance at 595 nm was measured on days 0, 1 and 3 of treatment.

### 4.4. Cell Cycle Analysis

Cell cycle analysis was performed as previously described [13,52]. Briefly, 1 × 10^5^ cells were seeded per well of a six-well plate, incubated overnight, then treated with either the vehicle or combinations of bac I and bac II for 24 h. Cells were resuspended in ice-cold Dulbecco’s phosphate buffered saline (DPBS) with drop-wise addition of 100% ice cold ethanol to a final concentration of 70% ethanol. Cells were centrifuged at 300× *g* for 5 min and cell pellets resuspended in 0.25% Triton X-100 in DPBS. Cells were stained for 2 h in darkness with 25 μg/mL propidium iodide (Sigma-Aldrich, St Louis, MO, USA) and 40 μg/mL bovine pancreas ribonuclease A (Sigma-Aldrich, St Louis, MO, USA) in DPBS and analysed on the BD FACSCanto II cell analyser (BD Biosciences, San Jose, CA, USA). Doublet populations were excluded and 50,000 single cell events were captured per sample. Data was analysed using FlowJo software v10.4.0 (FlowJo, LLC, Ashland, OR, USA), specifically by applying the Watson model to determine the percentage of cells in each stage of the cell cycle.

### 4.5. Apoptosis Assay

Apoptosis was determined by annexin-V and propidium iodide (PI) staining, using the Annexin-V-FLUOS staining kit (Roche Diagnostics, Mannheim, Germany), as previously described [14]. Briefly, cells were seeded at 1 × 10^5^ per well of a six well plate, cultured overnight then treated with either vehicle or bac I and bac II for 24 h. Cells were harvested and stained with annexin-V and propidium iodide for 15 min in darkness at room temperature then run on a BD FACSCanto II cell analyser (BD Biosciences, San Jose, CA, USA). Compensation was performed to account for the spectral overlap between fluorochromes. Cells were gated to exclude debris and doublets and at least 10,000 single cell events were acquired per sample. Data was analysed using FlowJo software v10.4.0 (FlowJo, LLC, Ashland, OR, USA).

### 4.6. Scratch Wound (Wound Closure) Migration Assay

The scratch wound assay was performed as described previously [13,53]. Cells were seeded in complete medium at 3 × 10^4^ (T47D and MCF7), 4 × 10^4^ (MDA-MB-231), or 5 × 10^4^ (BT-474) cells per well of a 96 well plate, incubated under standard conditions until 80–90% confluent, then serum starved overnight. A circular wound was made with a p10 pipette tip. Cells were resuspended in complete medium supplemented with 1 μg/mL mitomycin C (Sigma-Aldrich, St Louis, MO, USA) to inhibit cell proliferation, and either vehicle or bac I and/or bac II. The area of wound closure was measured using NIS Elements software (Nikon, Tokyo, Japan).

### 4.7. Spheroid Invasion Assay

The Cultrex**^®^** 3D culture spheroid cell invasion assay (Trevigen, Gaithersburg, MD, USA) was set up as per the manufacturer’s instructions using the kit contents, including spheroid formation extracellular matrix (ECM) and invasion matrix. MDA-MB-231 cells were seeded in the spheroid formation ECM at 3000 cells per well of a Costar**^®^** 96-well ultra-low attachment plate (Corning, NY, USA) followed by incubation for 72 h. Spheroids were subsequently embedded in invasion matrix and vehicle or drug treatments were added. The area of invasion on day 3 was measured using NIS elements software (Nikon, Tokyo, Japan) and was expressed normalised to the mean invasion of the vehicle control.

### 4.8. Analysis of Aquaporin-1 Expression by Quantitative PCR

Breast cancer cell lines were seeded at 3 × 10^5^ cells per well in six-well plates and incubated until 70 to 80% confluent. RNA was isolated with the PureLink RNA mini kit (Life Technologies, Carlsbad, CA, USA) at time 0 and following 24 h treatment with vehicle or combinations of bac I and bac II. The iScript cDNA Synthesis Kit (Bio-Rad Laboratories, Hercules, CA, USA) was used to reverse transcribe 200 ng of RNA. Transcript expression was determined as previously described [14], using multiplex TaqMan Gene Expression Assays for aquaporin-1 (AQP1; Hs01028916_m1; Applied Biosystems, Foster City, CA, USA) as the target gene, and coiled-coil serine rich protein 2 (CCSER2; Hs00982799_mH; Applied Biosystems, Foster City, CA, USA) as the reference gene. Reactions were performed using the Applied Biosystems ViiA 7 Real Time PCR System (Life Technologies, Carlsbad, CA, USA) with activation for 30 s at 95 °C, followed by 40 cycles of 15 s at 95 °C, and 30 s at 60 °C. Results were calculated using the 2^−ΔCt^ relative quantification method, normalizing to CCSER2 reference gene.

### 4.9. Statistical Analysis

Data were analysed using GraphPad Prism version 7.0c for Mac OS X (GraphPad Software, La Jolla, CA, USA) and is presented as means ± standard deviation (SD). Statistical analysis was done by one-way analysis of variance (ANOVA) with Bonferroni post-hoc test unless otherwise stated.

## 5. Conclusions

This study is the first of its kind to test the anti-cancer potential of bac I and bac II combined on breast cancer cell lines from each of the three molecular subtypes. Bac I and bac II combined behaved synergistically, the combination was more potent than either drug alone, and all subtypes were susceptible to their effects; however, the degree of susceptibility was cell line-specific. In particular, the more aggressive TNBC (MDA-MB-231) and HER2+ (BT-474) cell lines were the most sensitive. Further studies will explore the toxicity and pharmacokinetics of the two drugs in mouse models of breast cancer and elucidate the exact mechanisms underlying the observed effects.

## Figures and Tables

**Figure 1 molecules-24-03539-f001:**
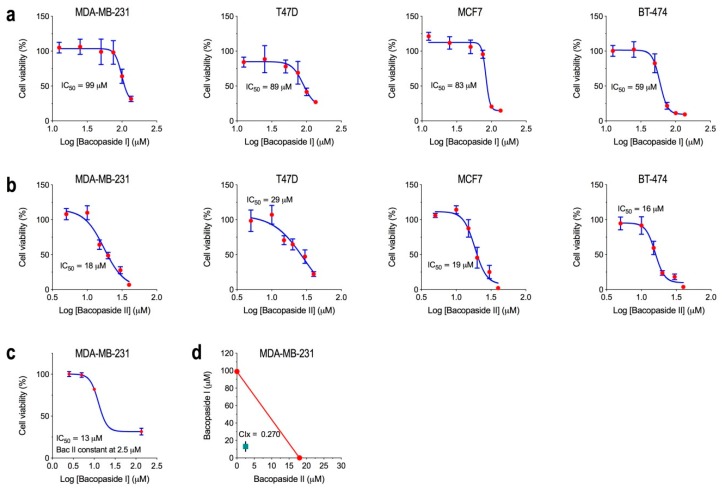
The IC_50_ for Bac I and Bac II in breast cancer cell lines. Breast cancer cell lines were treated with different doses of bac I or bac II and their cell viability determined by MTS assay. Data are the mean ± SD of six replicates from a representative experiment normalised to the vehicle control for bac I (**a**) and bac II (**b**). Non-linear regression analysis for bac I and bac II were used to calculate IC50 values for each drug. Relationships to the vehicle control were determined by one-way ANOVA with Bonferroni test. Regarding the synergy of bac I and bac II for MDA-MB-231 (**c**), cells were treated with varying doses of bac I combined with a constant dose of bac II (2.5 μM) for 24 h. Data are the mean ± SD of six replicates normalised to the vehicle control, with IC50 determined by non-linear regression analysis. The IC50s for each drug used individually (**d**) were plotted as an isobologram (red theoretical line) for MDA-MB-231. The IC50 combination of bac I and II was also plotted, with the 95% CI represented by the black line.

**Figure 2 molecules-24-03539-f002:**
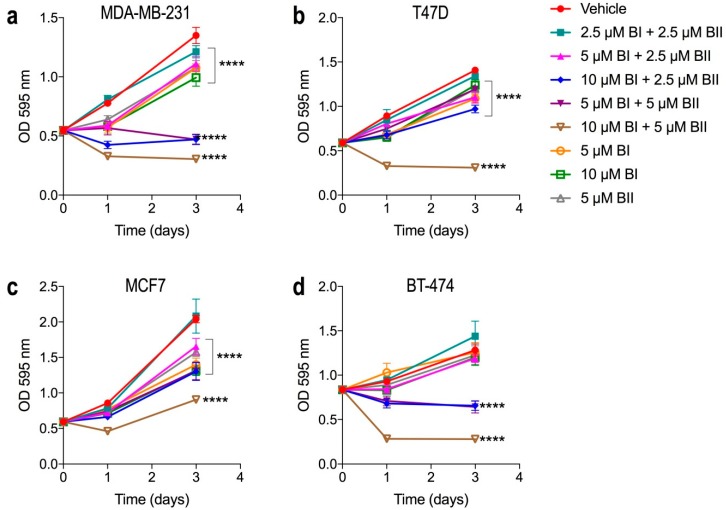
Bac I and II combined reduced breast cancer cell proliferation. BI: bac I; BII: bac II. Cells were seeded in 96-well plates and treated with bac I (BI) and bac II (BII), alone or in combination: **a**, MDA-MB-231; **b**, T47D; **c**, MCF7; **d**, BT-474. Plates were stained with crystal violet on days 0, 1, and 3 of treatment. Data are the mean ± SD of six replicates from a reproducible experiment. *p*-values shown are for absorbance on day 3, comparing treatment relative to the vehicle, as determined by two-way ANOVA with multiple comparisons and Bonferroni test (**** *p* < 0.0001).

**Figure 3 molecules-24-03539-f003:**
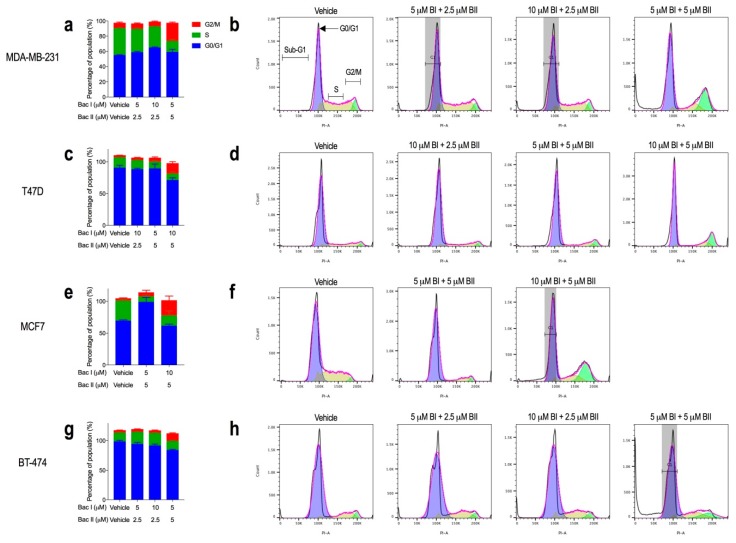
Higher combinations of bac I and II induced G2/M arrest in breast cancer cell lines. Cells were treated with vehicle or combinations of bac I and bac II for 24 h. Cells were stained with propidium iodide (25 μg/mL) and bovine pancreas ribonuclease A (40 μg/mL), and analysed on the BD FACSCanto II cell analyser. The percentage of the population in G0/G1, S, and G2/M phases of the cell cycle (left) and representative histograms (right) of propidium iodide-stained DNA, also showing sub-G1 events, for MDA-MB-231 (**a**, **b**), T47D (**c**, **d**), MCF7 (**e**, **f**) and BT-474 (**g**, **h**) are shown. Data are the mean ± SD of three biological replicates. Two-way ANOVA with multiple comparisons and Bonferroni test was performed for statistical analysis. Grey-shaded regions represent the constraints applied to improve the fit of the Watson model.

**Figure 4 molecules-24-03539-f004:**
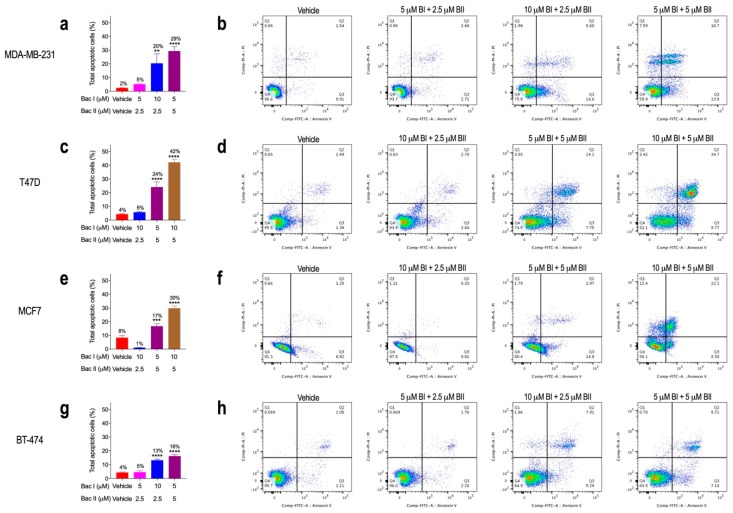
Higher-dose combinations of bac I and II induced apoptosis in breast cancer cell lines. Graphs (**a**,**c**,**e**,**g**) show the effect of the vehicle or bac I and bac II combined on the percentage of total apoptotic cells (early and late), with the representative scatterplots (**b**,**d**,**f**,**h**) showing the population gates of viable cells (left lower quadrant) or cells in early apoptosis (right lower quadrant), late apoptosis (right upper quadrant), or necrosis (left upper quadrant). Results for MDA-MB-231 (**a**,**b**), T47D (**c**,**d**), MCF7 (**e**,**f**) and BT-474 (**g**,**h**) are the mean ± SD of apoptotic cells from a representative experiment (n = 3). *p*-values, relative to vehicle, were determined by one-way ANOVA with Bonferroni test (** *p* = 0.001; *** *p* = 0.0003; **** *p* < 0.0001).

**Figure 5 molecules-24-03539-f005:**
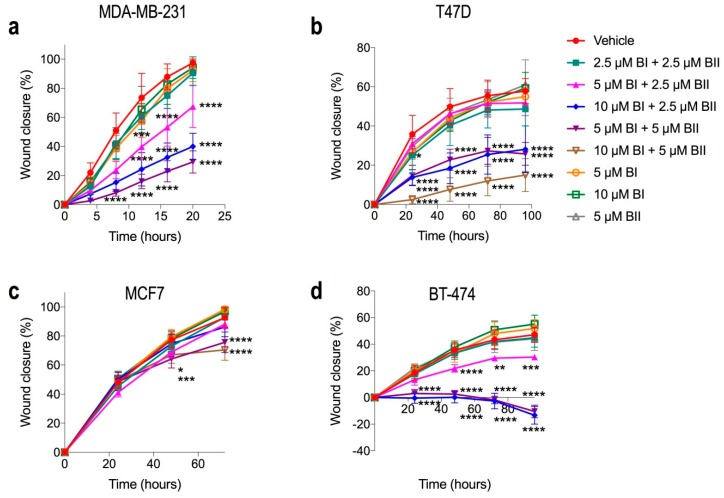
Cytotoxic and non-cytotoxic combinations of bac I and II reduced breast cancer cell migration. (**a**–**d**) The percentage of wound closure relative to closure at time 0, for MDA-MB-231 (**a**), T47D (**b**), MCF7 (**c**) and BT-474 (**d**). Wound area was measured using NIS-Elements software (Nikon, Tokyo, Japan). Data are the mean ± SD of six replicates from a representative experiment repeated in triplicate. P-values are for closure relative to vehicle at each time point, determined by one-way ANOVA with Bonferroni test (* *p* ≤ 0.045; ** *p* = 0.003; *** *p* ≤ 0.0007, **** *p* < 0.0001).

**Figure 6 molecules-24-03539-f006:**
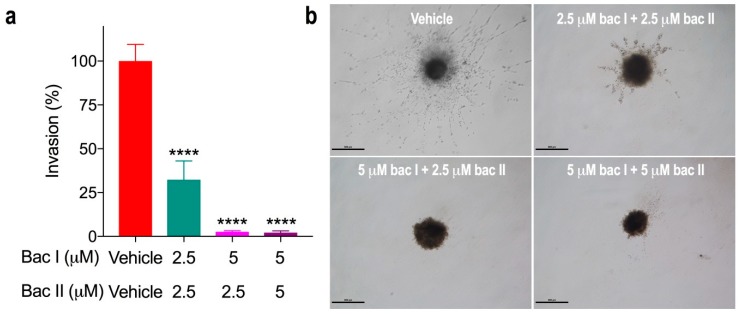
Non-cytotoxic combinations of bac I and II reduced MDA-MB-231 invasion in the spheroid invasion assay. MDA-MB-231 cells formed spheroids for three days prior to being embedded in the invasion matrix and treated with the vehicle, or bac I and bac II. The size of the spheroid was measured using NIS-Elements software (Nikon) daily for up to 4 days. (**a**) Invasion on day 3 of treatment was normalised to the average vehicle and expressed as the mean ± SD of three replicate wells. P-values are relative to the vehicle **** *p* < 0.0001). (**b**) Representative microscopic images (40× magnification; scale bar = 500 μm).

**Figure 7 molecules-24-03539-f007:**
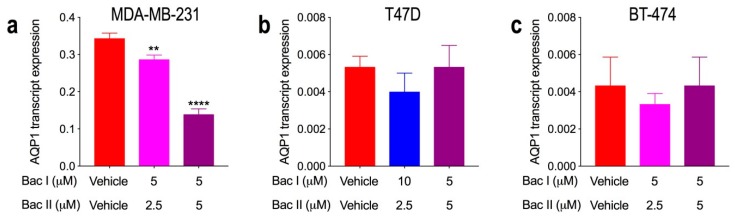
Relative aquaporin-1 (AQP1) transcript expression for breast cancer cell lines and the effect of bac I and II combined. Cells were treated for 24 h with the vehicle or bac I and II. Effect of bac I and II on transcript expression of AQP1 relative to vehicle for (**a**) MDA-MB-231 (** *p* = 0.004, **** *p* < 0.0001), (**b**) T47D, and (**c**) BT-474. Data are the mean ± SD of three replicates, with AQP1 transcript expression normalised to the CCSER2 reference gene and expressed as 2^−ΔCt^. P-values are relative to the vehicle and determined by one-way ANOVA with Bonferroni test.

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
