# Peer review of "Bacopasides I and II Act in Synergy to Inhibit the Growth, Migration and Invasion of Breast Cancer Cell Lines"

_molecules, 2019, doi:10.3390/molecules24193539_

Round 1
Reviewer 1 Report
The manuscript investigates the strategy of drug combinations to target breast cancer and have utilized four cell lines that covers the types of breast cancer know. The efforts taken by the Authors to present the manuscript is appreciated. The manuscript is well written and provides ample information and background literatures to substantiate the use of bac I and II.
It is however necessary to correlate the findings of the study’s translational aspect in the manuscript. Can the dose of uM used in vitro be reproducible In in vivo experiments. I recommend discussing the translational aspects in the manuscript which will strengthen the research study.
Author Response
Response: We have added the following paragraph in the discussion to address the reviewer's comments re translation of the in vitro doses to human clinical trials.
In terms of translation to human trials, based on the guide recommended by the U.S. department of health and human services, food and drug administration (FDA)[47], the estimation of the maximum safe starting dose in initial clinical trials for bac I and bac II in adults would be doses of 0.5 and 0.2 mg/kg, respectively. Hence, an average 70 kg human would receive a dose of Bac I of approximately 20 mg/m2 and for Bac II 7.5 mg/m2. These doses are much lower in comparison with one of the common metastatic breast cancer chemotherapy agents, paclitaxel, with the recommended dose of 175 mg/m2.
Reviewer 2 Report
The work is well written and easily readable. The experiments are generally easy to interpret and make sense.However, the whole molecular part is missing to explain the physiological effects observed on the cells.
The authors could use a non-tumor cell line to demonstrate non-toxicity of Bacopaside (eg MCF-10-A normal breast cell line).
In Figure 2 are shown picture of apoptotic cells by light microscope but only with a scanning electron microscope (SEM) it is possible conclude that cells are in apoptosis. The authors could shift this figure as supplementary and add another experiment for apoptosis (as Annexin assay in figure 4)
The authors do not evaluate any apoptotic pathways, the authors should do an mRNA/protein analysis of Bcl2 and Bax or capspase 3 and 9 on at least one cell line.
In Figure 3 the Cell proliferation assay is shown. Why is it measured by criystal violet assay? MTT assay is the most suitable technique otherwise cell viability is measured.
In the wound healing assay figure the authors should add at least one photo of the control and treatment per cell line tested (also as an additional file)
The authors should explain which is the molecular mechanism of the invasion inhibition. The Authors should investigated the protein expression and activity of MMPs (at least MMP2 and MMP9) as the authors conclude in the 342-344 line of Discussion.
Reviewer 3 Report
Dear Authors,
I consider you manuscript interesting and with a lot of data to furthers suport the use of the described substances. There are some minor points I think to be reviewed:
Line 36, please could this reference be updated with more recente one?
Line 37, I think breast cancer can be classified according to the receptors expressed, but there are histological classification too. Please, change sentence.
Line 63, missing parenthesis
Line 64. In the abstract, it is explained what bac I and bac II are, but not here in the main text.
Please, check abastrcat as well, the frst line Bacopaside (bac) I and bac II...shouldn´t be Bacopaside (bac) I and II are triterpene saponins purified from the medicinal herb 15 Bacopa monnieri.
Author Response
Line 36, please could this reference be updated with more recent one?
Response: we have inserted a more recent reference (2018)
Line 37, I think breast cancer can be classified according to the receptors expressed, but there are histological classification too. Please, change sentence.
Response: We have rewritten the sentence "Breast cancer has been classified traditionally according to characteristics such as tumour size, lymph node involvement, histological grade and the receptors expressed"
Line 63, missing parenthesis
Response: We have inserted parentheses: (reviewed in [16])
Line 64. In the abstract, it is explained what bac I and bac II are, but not here in the main text.
Response: We have inserted the following (lines 62-63) "and no studies have been performed using the purified compounds bacopaside (bac) I or II."
Please, check abastrcat as well, the frst line Bacopaside (bac) I and bac II...shouldn´t be Bacopaside (bac) I and II are triterpene saponins purified from the medicinal herb 15 Bacopa monnieri.
Response: We have corrected this in first line of abstract.
Round 2
Reviewer 2 Report
None